# The Localized Corrosion and Stress Corrosion Cracking of a 6005A-T6 Extrusion Profile

**DOI:** 10.3390/ma14174924

**Published:** 2021-08-30

**Authors:** Jijun Ma, Jing Sun, Quanmei Guan, Qingwei Yang, Jian Tang, Chengxiong Zou, Jun Wang, Bin Tang, Hongchao Kou, Haisheng Wang, Jun Gao, Jinshan Li, William Yi Wang

**Affiliations:** 1CRRC Tangshan Co., Ltd., Tangshan 063035, China; sjc-sunjing@tangche.com (J.S.); sjc-guanquanmei@tangche.com (Q.G.); wanghaisheng@tangche.com (H.W.); sjc-gaojun@tangche.com (J.G.); 2State Key Laboratory of Solidification Processing, School of Materials Science and Engineering, Northwestern Polytechnical University, Xi’an 710072, China; 1351724745@mail.nwpu.edu.cn (Q.Y.); nwputj@mail.nwpu.edu.cn (J.T.); zcx2016@mail.nwpu.edu.cn (C.Z.); nwpuwj@nwpu.edu.cn (J.W.); toby@nwpu.edu.cn (B.T.); hchkou@nwpu.edu.cn (H.K.); 3Innovation Center, NPU Chongqing, Chongqing 401135, China

**Keywords:** stress corrosion cracking, pitting, intergranular corrosion, grain boundaries

## Abstract

In the present work, the localized corrosion and stress corrosion cracking (SCC) behaviors of a commercial 6005A-T6 aluminum extrusion profile was studied comprehensively. The velocity of crack growth in self-stressed double-cantilever beam (DCB) specimens under constant displacement was estimated, which also provides insight into the local microstructure evolutions at the crack tips caused by the localized pitting corrosion, intergranular corrosion (IGC), and intergranular SCC. Characterizations of local corrosion along the cracking path for a period of exposure to 3.5% NaCl were revealed via optical microscope (OM), scanning electron microscope (SEM), and electron backscatter diffraction (EBSD). The typical features of the pits dominated by the distribution of precipitates included the peripheral dissolution of the Al matrix, channeling corrosion, intergranular attack, and large pits in the grains. The discontinuous cracking at the crack tips indicated the hydrogen-embrittlement-mediated mechanism. Moreover, the local regions enriched with Mg_2_Si and Mg_5_Si_6_ phases and with low-angle grain boundaries presented better SCC resistance than those of the matrix with high-angle grain boundaries, supporting a strategy to develop advanced Al–Mg–Si alloys via interfacial engineering.

## 1. Introduction

With the quick development of the high-speed railway and the operation of the China Railway High-speed service for more than a decade, one of the greatest challenges is the management/maintenance of these trains in environmental conditions [1,2,3]. In order to support the foundations for interactive corrosion risk management, it is essential to reveal the foundations of environmentally induced cracking caused by corrosion [4,5]. Al–Mg–Si–(Cu) alloys (6XXX series) are generally considered to be one kind of important lightweight structural metal material, and have been widely applied in the aerospace, transport, automotive, and shipbuilding industries [1]. In line with the well-established heat treatment, the decomposition sequence of the solid solution of Al–Mg–Si alloys can be expressed as [6,7,8]:SSSS → solute clusters → GP Zones → β″ → β′ →β
where SSSS represents the supersaturated solid solution. Through increasing the concentration of Cu in Al–Mg–Si alloys, their strength can be improved [9], and the decomposition sequence can be changed by enhancing the number of possible phases and the complexity of the precipitation sequence [9,10,11]. In particular, the precipitation hardening phases in Al–Mg–Si alloys are β-Mg_2_Si and λ-Al_5_Cu_2_Mg_8_Si_6_, presenting excellent resistance to exfoliation and SCC [9]. The addition of Cu increases the intergranular corrosion (IGC) susceptibility, which can be reduced by over-aging to a T7-type temperature [9]. Moreover, the size and the morphology of precipitates play important roles in improving both the strength and the corrosion resistance of Al alloys [9,12,13,14,15]. The coarse particles have a direct effect on formability, and also generally play a cathodic role owing to the higher concentrations of Cu, Fe, Mn, and Si than of the Al matrix [14]. The most typical structures of 6XXX alloys are the plate-like β-Al_5_FeSi particles and the rounded α-Al_12_(Fe,Mn)_3_Si particles [16]. The brittle, monoclinic β-Al_5_FeSi phase, which is insoluble during solution heat treatment, is associated with poor workability and causes a poor surface finish [16]. The presence of coarse, elongated particles is the key microstructural feature affecting the fracture behavior of 6XXX Al [16]. The detrimental elongated β-type particles are transformed into rounded α-type particles by heat treatment [16]. On the other hand, in order to enhance corrosion resistance, it is beneficial to produce very fine precipitates [12].

Under the stress and the corrosion environment, the stress corrosion cracking (SCC) of 6XXX alloys has been captured and investigated widely [17,18,19,20]. It is essential to comprehensively investigate the SCC mechanisms, including local anodic dissolution (LAD) of the solute-free zones or the GB precipitates, hydrogen-induced embrittlement (HIE), hydrogen-enhanced decohesion (HEDE), hydrogen-enhanced localized plasticity (HELP), hydrogen-enhanced vacancy stabilization mechanism (VM) [9,21,22,23,24], etc. Based on the superdislocation model, it is worth mentioning that dislocation emanating from crack tips greatly enhances local stresses a few nanometers ahead of the crack tip, causing significant hydrogen accumulation, resulting in HEDE [22]. It has been found that hydrogen has a strong interaction with Al vacancies [25] and reduces the stacking fault energy, affecting the nucleation of dislocations [24]. In the view of bonding charge density [4,26,27], the notable accumulation of bonding electrons of interstitial atoms occupying either the T-site or the O-site yields a dramatic reduction in the Al–X and Al–Al bonding charge density and bonding strength, supporting atomic and electronic insights into HIE [4]. The energetically favorable T-site occupation has been estimated [4], matching well with the previous theoretical observations [25,26].

In the present work, the localized corrosion and stress corrosion cracking behaviors of a commercial 6005A-T6 aluminum extrusion profile were studied comprehensively to reveal the key SCC mechanisms. The velocity of crack growth in self-stressed DCB specimens under constant displacement was estimated, also providing insight into the local microstructural evolutions at the crack tips caused by the localized pitting corrosion, intergranular corrosion (IGC), and intergranular SCC (IGSCC). The contributions of precipitation, high-angle grain boundaries (HAGBs), low-angle grain boundaries (LAGBs), and the hydrogen-assisted cracking were experimentally and theoretically evaluated, paving a path to the development of advanced, SCC-resistant Al–Mg–Si alloys.

## 2. Materials and Methods

The commercial 6005A-T6 aluminum extrusion profile for the frames of high-speed trains—the composition and the schematic diagram of which are presented in Table 1 and Figure 1a, respectively—was utilized in the present study. Based on our previous works [1,2], it is understood that pitting and intergranular corrosion along the extrusion direction would be the most serious. Thus, it would be essential to the estimate/investigate the SCC resistance along the deformation direction. Based on the national and international standards “GB/T 12455.1-1990 High-Strength Alloys—Method of Stress Corrosion Test for Double Cantilever Beam (DCB)” and “ISO/DIS 7539-6:2018 Corrosion of Metals and Alloys—Stress Corrosion Cracking—Part 6: Preparation and Use of Pre-cracked Specimens for Tests under Constant Load or Constant Displacement”, the pre-cracked self-stressed DCB specimens along the extrusion direction were fabricated, the schematic diagrams of which are presented in Figure 1a,b-1. These two free surfaces are referred to as A-Plane and B-Plane, respectively. After the SCC test, the DCB specimen was cut at the quarter positions referring to the surfaces, which are referred to as 0.25B plane and 0.75B plane, respectively.

### 2.1. Corrosion Test

With the guidance of these aforementioned national and international standards, the 6005A-T6 alloy DCB specimens’ geometry is presented in Figure 1a. The main crack length was obtained by setting a constant displacement of 3~6 mm, utilizing the bolts to spread apart the two arms of the pre-cracked self-stressed DCB specimen in order to stress it without requiring any other equipment. During the test, the pre-cracked DCB specimens were immersed in 3.5% NaCl solution with a pH of ~6–7 at 35 ± 1 °C. Moreover, such stress still permitted quantification of the stress intensity at the root of the crack [27]. The wetting process in air was performed through utilizing a small pump, making the 3.5% NaCl solution circulate in the testing container. It is worth mentioning that the solution should be replaced weekly. Figure 1b shows the specimen before and after the SCC test, together with the enlarged region with the propagated cracks. Based on the measured crack length at different times, the crack propagation rate (ϑ=da/dt) and the stress intensity factor (K) can be obtained. Initially, the crack length was measured intensively every few hours, since the da/dt was very fast within this so-called stage I, in which the logϑ increased linearly and rapidly with increasing K [27]. Afterwards, referring to the ϑ−K characterization, a saturation level was reached, which is called stage II, yielding the ϑ independent of K. Finally, the ϑ became dependent on K, which is referred to as stage III. When ϑ ≤ 10^−9^ mm/s, the SCC test was stopped. It is worth mentioning that the crack growth rate is generally estimated from sequential measurements averaged on both sides of the DCB specimen [28]. Therefore, both sides should be characterized spontaneously.

### 2.2. Microstructure Characterizations

During the SCC tests, the DCB specimens were cleaned in anhydrous ethanol for 30 min using an ultrasonic cleaning machine, which was utilized to measure the crack length via an OLYMPUS GX51F optical microscope (Olympus Cop., Tokyo, Japan). Both the surface and the cross-sections after polishing were characterized via scanning electron microscopy (SEM) and electron backscattered diffraction (EBSD), the latter of which was also applied to qualitatively and quantitatively analyze the grain boundary misorientation, phases, and their networks. 

## 3. Results and Discussion

### 3.1. The Crack Growth of the Pre-Cracked Self-Stressed DCB Specimen

Figure 2 and Figure 3 display the crack propagations in the A-plane and the B-plane, respectively, of the pre-cracked self-stressed DCB specimen characterized by OM. There are several advantages of the pre-cracked DCB specimens [27]: Firstly, they obviate the need to wait for a corrosion pit to grow, saving time. Secondly, they enable one to avoid an erroneous conclusion of immunity to SCC because of a non-pitting combination of alloy and environment. Thirdly, they enable one to obtain a conservative evaluation by evaluating the material in the presence of the ultimate flaw—namely, a sharp crack. Fourthly, if the specimen meets certain criteria, the methods of fracture mechanics can be used to predict, from the behavior observed in one geometry of specimen and crack, what will happen in other geometries. Accordingly, the measured steady-state crack propagation rate at stage II of the A and B surfaces is 5.88 × 10^−7^ mm/s and 6.09 × 10^−7^ mm/s, respectively, while the stress intensity factor of SCC (K_ISCC_) is 177.9 MPa m^1/2^ and 85.4 MPa m^1/2^, respectively.

It should be noted that there are some significant differences in the crack length and the morphology between these two surfaces, suggesting the microstructure-dependent IGSCC rate. Based on our previous investigations [2], it is understood that corrosion along the direction of distribution of constituent precipitations and elongated grains yields serious IGC, which could contribute to the formation of various IGSCC morphologies along different grains. Therefore, it is essential to comprehensively characterize the localized corrosion of the pre-cracked self-stressed DCB specimen in detail.

### 3.2. Characterizations of the Localized Corrosion and SCC Microstructures

Figure 4 displays the localized corrosion and SCC microstructures by SEM. The discontinuous cracks in the SCC propagation paths together with pits and precipitations are presented in Figure 4a–c. Interestingly, there are some subcracks at the crack tips. Moreover, several precipitates are located at the crack paths. Based on the characterizations and analysis of the crack tips shown in Figure 4d–f, the pitting and the IGC are also captured. We found that specimens that were strengthened by large Guinier–Preston (G–P) zones exhibited coarse slip characteristics and proved to crack readily, while specimens hardened by small G–P zones exhibited fine slip and cracked slowly [27]. Similarly, according to the anodic-dissolution-based SCC mechanism, higher cracking rates of AA5083 alloys can be expected at the positions with more β-phase on grain boundaries and the more continuous and interconnected β-phase [29]. Grain boundary regions not covered by β-phase films give rise to a retarding effect, wherein the crack must jump the resisting region composed of alloy matrix ligaments, leading to slower cracking with more apparent scattering behavior [29].

As presented in Figure 5, the classical cracking features at the IGSCC tips were characterized via EBSD. The electron image of the crack tips together with precipitates and IGC in Figure 5a was utilized to highlight the selected locations for further characterizations. In particular, the characterizations of selected regions in the views of band contrast, Euler color, the inverse pole figure (IPF) of the x-, y-, and z-axes, and the phase map are displayed in Figure 5b–e, respectively. It can be found that the dominated cracking path crosses through several grains, presenting the typical feature of transgranular SCC in the precipitation-free zone (PFZ). Moreover, the angle between the cracking path and the extrusion direction (the elongated grains) was almost 45°, which follows Schmid’s law, presenting the maximum shear stress along this slip plane.

Figure 6 highlights the classical cracking features at the IGSCC tips along the elongated grains. It can be seen that the cracks grow along the elongated grains located at the PFZs. On the other hand, those grain boundaries (GBs) enriched with the Mg_5_Si_6_ and Mg_2_Si phases present a good resistance to the SCC, although they segregate at the GBs parallel to the main cracking path. Based on our previous investigations of the local corrosion features of the 6005A-T6 alloy [2], we found that the Mg/Si ratio is a critical indicator in the development of novel, advanced, corrosion-resistant Al alloys, since the Si–Al interface and primary α-Al matrix remained unattached, presenting much better IGC resistance than that of the 5083-H111 and 6082-T6 alloys. In general, the Mg_2_Si particles are anodic to the matrix, yielding local corrosion on their surface, while the Si particles are cathodic to the matrix [19]. The formation of solute clusters in 6XXX alloys—including Mg, Si, and Mg–Si—could also be obtained during various heat treatment conditions [11]. During the corrosion, the preferential dissolution of Mg and the enrichment of Si make Mg_2_Si transform from anode to cathode, leading to the anodic dissolution and corrosion of the alloy base at its adjacent periphery at a later stage [19]. 

However, during the SCC tests, it is suggested that the effects of those aforementioned solute clusters and the main intermetallic precipitations on the corrosion behavior should be double checked. In order to improve the SCC resistance of Al alloys, their contributions to the enhancements of strength and corrosion resistance should be considered simultaneously [1]. On the one hand, the formation of Mg_2_Si improves the formability and strength; the supersaturated Si forming the Q (AlCuMgSi) and the unsolvable Al_6_(FeMnSi) phases decreases the concentrations of solutes around the grain boundary, forming precipitation-free zones (PFZs), all of which cause serious IGC [2]. On the other hand, those solutes and precipitations also cause the solid solution and the precipitation-strengthening effects, which would be beneficial to improve the SCC resistance under stressful and corrosive conditions by improving the mechanical performance. 

### 3.3. Pit-to-Crack Transformations and IGSCC

Figure 7 displays the localized corrosion and SCC microstructures inside the pre-cracked self-stressed DCB specimen. We found that the multi-cracking paths occur at the PFZ zones, which are parallel to the extrusion directions. At the cracking tips, both the transgranular and intergranular SCC can be captured, indicating the serious IGC inside the DCB specimen, as shown in Figure 7a. While these multiple cracks and branches obviously display the boundaries of some elongated grains, the stored dislocations during extrusion and the subgrains would contribute to those fine-grain zones enduring severe corrosion. Moreover, there are a great number of pits along the GBs of elongated grains, as shown in Figure 7b. In particular, the precipitates in the center of the pits highlight the contributions of the pits and IGC to the growth of the cracking path and IGSCC, as well as indicating the pit-to-crack transformation. Correspondingly, the typical morphologies of the localized corrosion around precipitates are presented in Figure 7c,d.

Furthermore, the pitting and IGSCC cracking features in the 0.75B plane were characterized by EBSD to reveal the fundamental reasons for the pit-to-crack transformation, as shown in Figure 8. We found that those main cracking paths are almost all along the PFZs, while the pits occur with a lot of discontinuous precipitates nearby. The heterogeneous structures composed of a series of hardening precipitates and solute clusters initiated localized corrosion at the secondary phases or their surrounding Al matrix, forming pits and beginning the degradation of the mechanical properties of the alloy [1]. The correlations between the microchemistry of grain boundary precipitates (GBPs) and SCC resistance have been widely investigated [13,14,30,31,32]. It is understood that precipitates usually have a composition different from that of the matrix, yielding a local chemistry change [1,33]. The corresponding electrochemical properties, including pitting corrosion and stress corrosion cracking, are detrimentally affected by PFZs [1,33]. In particular, some solute atoms could increase the corrosion resistance of Al–Mg alloys by promoting the formation of competitive GBPs and retarding the growth kinetics of the β-phase, resulting in better IGSCC resistance than those alloys consisting primarily of the β-phase [32]. Therefore, GB engineering would be an efficient approach, as it not only improves the mechanical properties of Al alloys, but also enhances their corrosion resistance.

### 3.4. Discussion

Based on those characterizations in terms of OM, SEM, and EBSD discussed above, it is apparent that GBs play an important role in the IGC, IGSCC, and SCC. Taking advantage of the Euler angles obtained via EBSD analysis, the GB misorientation statistics can be completed conveniently [2,34]. As presented in Figure 9, the statistics of high-angle grain boundaries (HAGBs) and low-angle grain boundaries (LAGBs) for the selected regions of the DCB specimen were compared. It can be seen that the main dominated crack growth path is along the HAGBs parallel to the extrusion direction, as shown in Figure 9a. At the cracking tips, it appears that there are a great number of precipitations segregating at the HAGBs, avoiding the crack propagation along the original main cracking path and resulting in failure to follow Schmid’s law, with the maximum shear stress along this slip plane, as displayed in Figure 9b. It is understood that the GBPs generally have a higher corrosion potential than the Al matrix, thereby acting as the cathode in the microgalvanic reaction [20]. The 6005A extrusion profiles with elevated copper content demonstrated IGC susceptibility due to the presence of GBPs that are cathodic relative to the PFZ [20]. On the other hand, the HAGBs match exactly the dominant cracking propagation in Figure 9c,d. While a great amount of LAGBs are located at the unattached regions, the networks of HAGBs hint at their significant contributions to IGC and IGSCC [2]. Similarly, it has been reported that there is a direct relationship between IGC and SCC, which has also been observed in 5XXX, 6XXX, and 7XXX alloys [20]. These observations suggest that the metallurgical and environmental conditions that promote IGC in 6005A extrusions also promote SCC [20].

It is worth mentioning that, to date, there is no universal acceptance of any single mechanism to explain the SCC of Al alloys, although many research efforts have been devoted to developing and proving a single explanation for SCC [20]. In order to comprehensively investigate these aforementioned SCC mechanisms—including the HIE, HEDE, and LAD of the solute-free zones or the GB precipitates—the atomic and electronic bases for the hydrogen-embrittlement-mediated mechanism (i.e., HIE) were revealed by integrating the first-principles calculations [4] and molecular dynamics simulations [35], as shown in Figure 10. Here, first-principles calculations [4] were used to measure the hydrogen-mediated embrittlement of Al, while molecular dynamics simulations [35] revealed the effects of hydrogen on the deformation behavior and crack propagation of Al, Al_98_H_2_, and Al_95_H_5_ under axial tension at a strain rate of 1 × 10^−9^ s^−1^. During the tensile plastic deformation, the crack propagation paths (two groups of slip planes in red) of Al follow Schmid’s law, yielding an angle between the tensile force and the slip plane of 45° [35]. Moreover, the dislocations—presented as colored lines different from the blue matrix in Figure 10a—are also parallel to these two dominant cracking paths, along which there is maximum shear stress in the slip planes. With the increasing concentration of H, the dislocation density of the Al matrix is reduced. However, the cracking distance and the thickness of slips are enhanced, indicating the HIE mechanism and the accelerated cracking rate [35]. By enlarging the atomic radius of H and reducing that of Al, the slip bands enriched with H atoms can be presented clearly, as shown in Figure 10b. Furthermore, the H atoms are frequently captured in each slip plane, as displayed in Figure 10c. In terms of the bonding charge density [4,26,27] of the Al–H system shown in Figure 10d, it can be seen that the strength of Al–Al bonds located at the first nearest neighbor site of the H atom is dramatically decreased due to the notable accumulation of bonding electrons around H atoms, providing atomic and electronic insights into the HIE.

## 4. Conclusions

(1)The localized corrosion and SCC behavior of a commercial 6005A-T6 aluminum extrusion profile were studied comprehensively and conveniently via the self-stressed DCB specimens under constant displacement in 3.5% NaCl solution with a pH of ~6–7 at 35 ± 1 °C. The measured steady-state crack propagation rates of the A and B surfaces were 5.88 × 10^−7^ mm/s and 6.09 × 10^−7^ mm/s, respectively, while the stress intensity factors of SCC (K_ISCC_) were 177.9 MPa m^1/2^ and 85.4 MPa m^1/2^, respectively;(2)We found that the corrosion along the direction of distribution of constituent precipitations and elongated grains yields serious IGC, which could contribute to the formation of various IGSCC morphologies along different grains. While a great amount of LAGBs are located at the unattached regions, the networks of HAGBs hint at their significant contributions to the IGC and IGSCC;(3)We suggest that the effects of solute clusters and the main intermetallic precipitations on the corrosion behavior should be double checked. The local regions enriched with Mg_2_Si and Mg_5_Si_6_ phases and with low-angle grain boundaries present better SCC resistance than that of the matrix with high-angle grain boundaries, supporting a strategy to develop advanced Al–Mg–Si alloys via interfacial engineering;(4)In terms of the bonding charge density of the Al–H system, we found that the strength of Al–Al bonds located at the first nearest neighbor site of the H atom is dramatically decreased due to the notable accumulation of bonding electrons around H atoms, providing both atomic and electronic insights into the HIE.

## Figures and Tables

**Figure 1 materials-14-04924-f001:**
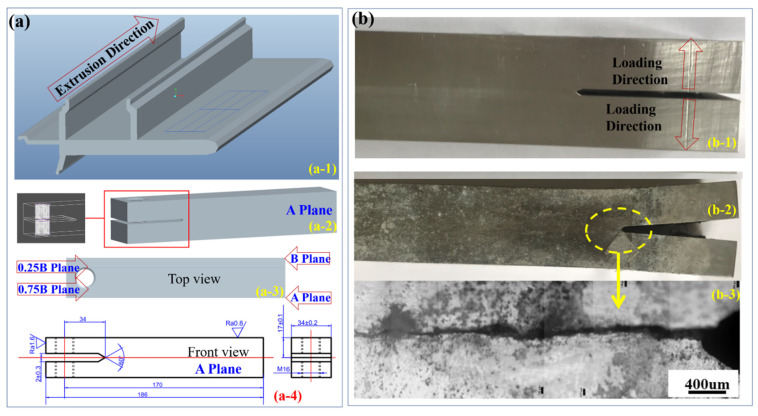
The fabrication of DCB specimens: (**a**) the schematic diagrams of the pre-cracked self-stressed DCB specimens along the extrusion direction; (**b**) the specimen before and after the SCC test together with the enlarged region with the propagated cracks. The additional numbers identify the various views of the DCB specimen. After the SCC test, the DCB specimen was cut at the quarter positions referring to the surfaces, which are referred to as 0.25B plane and 0.75B plane, respectively.

**Figure 2 materials-14-04924-f002:**
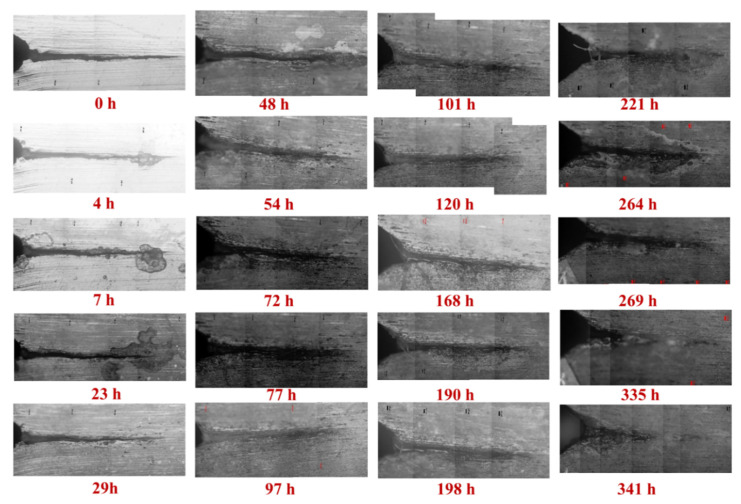
The crack propagations in the A-plane of the pre-cracked self-stressed DCB specimen, characterized by OM.

**Figure 3 materials-14-04924-f003:**
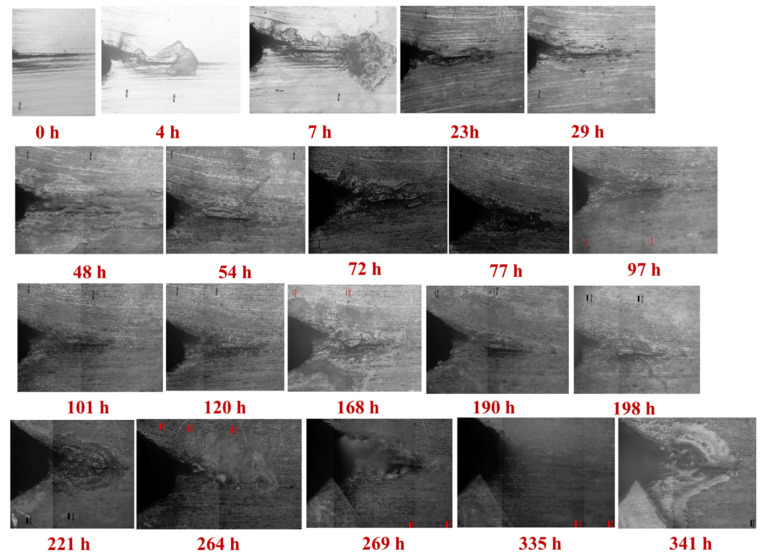
The crack propagations in the B-plane of the pre-cracked self-stressed DCB specimen, characterized by OM.

**Figure 4 materials-14-04924-f004:**
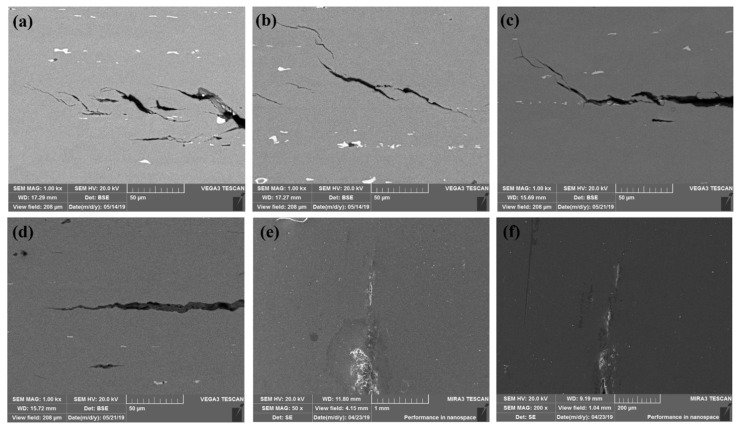
SEM images of the localized corrosion and SCC microstructures: (**a**–**c**) the discontinuous cracks in the SCC propagation paths together with pits and precipitations; (**d**) the crack tip; (**e**,**f**) the pitting and the IGC.

**Figure 5 materials-14-04924-f005:**
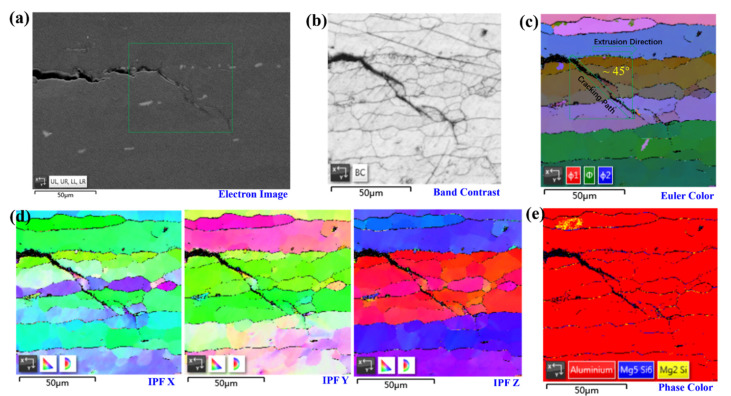
The classical cracking features at the SCC tips characterized by EBSD: (**a**) the electron image of the crack tips together with precipitates and IGC; (**b**–**e**) the characterizations of selected regions in terms of band contrast, Euler color, the inverse pole figure (IPF) of the x-, y-, and z-axes, and the phase map, respectively.

**Figure 6 materials-14-04924-f006:**
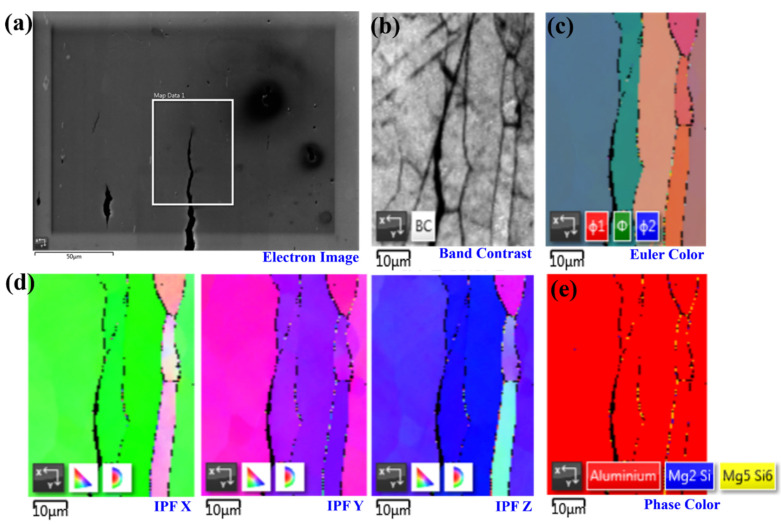
The classical cracking features at the IGSCC tips along the elongated grains, as characterized by EBSD: (**a**) the electron image of the crack tips together with precipitates and pits; (**b**–**e**) the characterizations of selected regions in terms of band contrast, Euler color, the inverse pole figure (IPF) of the x-, y-, and z-axes, and the phase map, respectively.

**Figure 7 materials-14-04924-f007:**
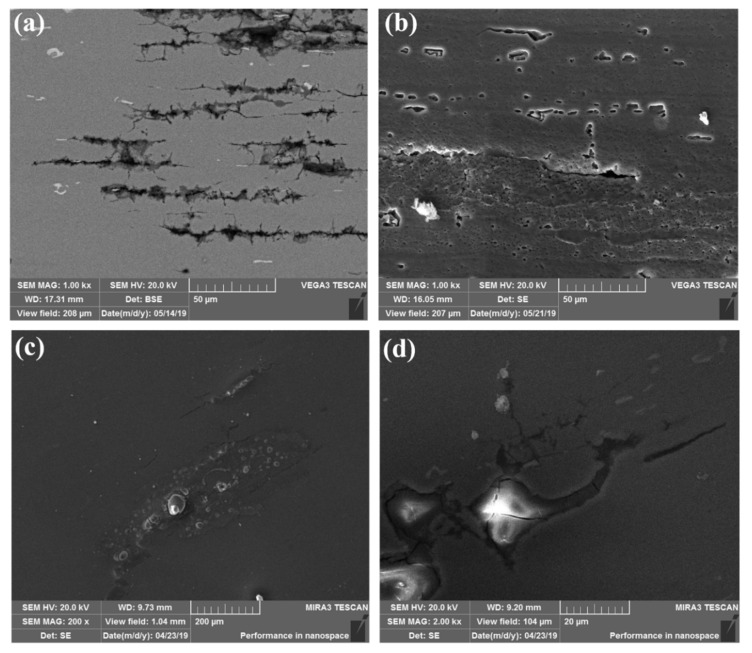
SEM images of the localized corrosion and SCC microstructures inside the pre-cracked self-stressed DCB specimen: (**a**) the multiple IGSCC cracks of the 0.25B plane; (**b**) the pitting and IGSCC of the 0.75B plane; (**c**,**d**) the typical morphologies of the localized corrosion around precipitates.

**Figure 8 materials-14-04924-f008:**
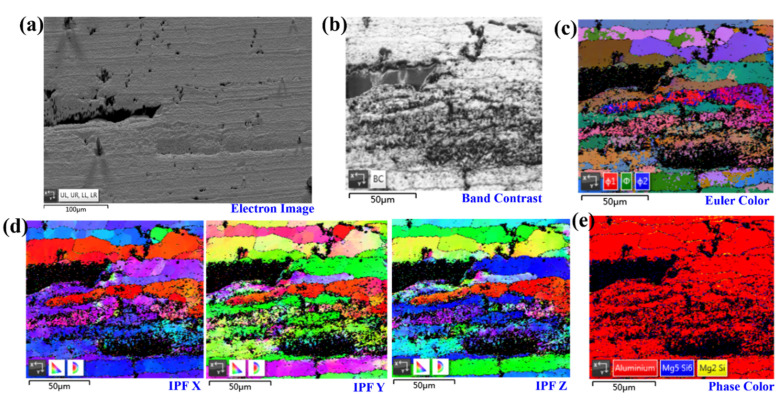
EBSD characterizations of the pitting and IGSCC features in the 0.75B plane: (**a**) the electron image of the crack tips together with precipitates and pits; (**b**–**e**) the characterizations of selected regions in terms of band contrast, Euler color, the inverse pole figure (IPF) of the x-, y-, and z-axes, and the phase map, respectively.

**Figure 9 materials-14-04924-f009:**
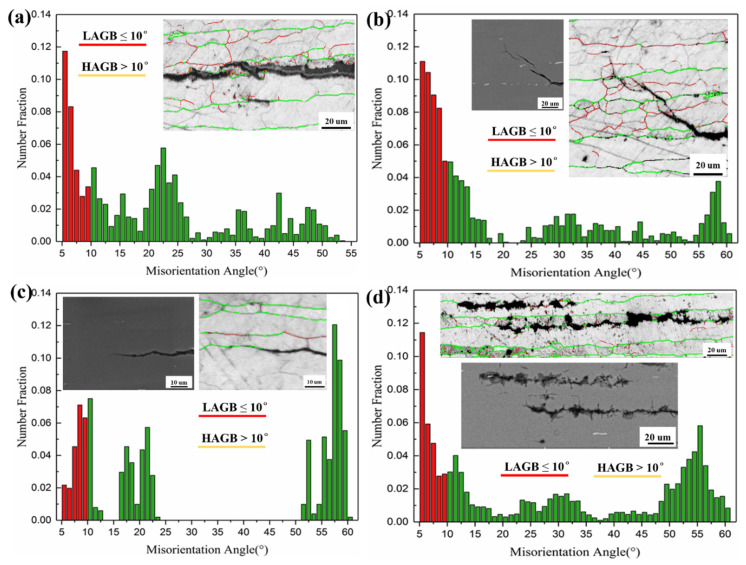
The statistics of high-angle grain boundaries (HAGBs) and low-angle grain boundaries (LAGBs) for the selected regions of the DCB specimen: (**a**) GB analysis along the crack growth path; (**b**) GB analysis of IGSCC; (**c**) GB analysis at a crack tip; (**d**) GB analysis of multiple cracks inside the DCB specimen.

**Figure 10 materials-14-04924-f010:**
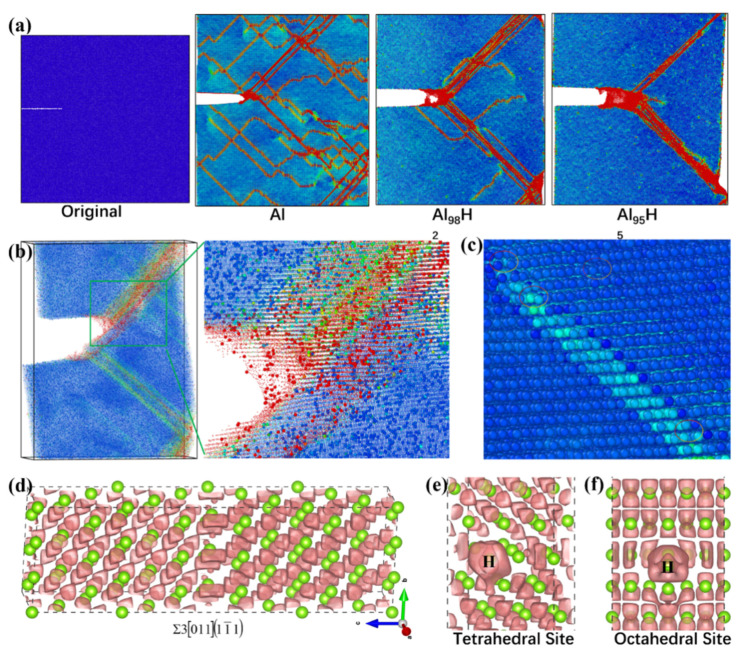
The atomic and electronic bases for the hydrogen-assisted cracking of Al [4,35]: (**a**) the effect of hydrogen on the deformation behavior and crack propagation of Al, Al_98_H_2_, and Al_95_H_5_ under axial tension at a strain rate of 1 × 10^−9^ s^−1^; (**b**) the trapped hydrogen-activated crack propagation and slip of Al_98_H_2_ presenting a clear contrast by enlarging the atomic radii of H atoms relative to those of Al atoms; (**c**) the trapped hydrogen at the tip of the dislocation; (**d**) the bonding charge density of $\sum 3\left[011\right]\left(1\bar{1}1\right)$ GB of Al; (**e**,**f**) hydrogen embrittlement for the tetrahedral and octahedral occupations, respectively, at the $\sum 3\left[011\right]\left(1\bar{1}1\right)$ GB. The von Mises strain in (**a**–**c**) was utilized to present the blue–green–red (BGR) gradient colors, with minimum and maximum values of 0.001 and 0.9, respectively.

**Table 1 materials-14-04924-t001:** Chemical composition of the commercial 6005A-T6 alloy at manufacture (wt %).

Elements	Si	Fe	Cu	Mn	Mg	Cr	Zn	Ti	Al
**Content**	0.5–0.9	≤0.35	≤0.30	≤0.30	0.40–0.7	≤0.30	≤0.20	≤0.10	Bal.

## Data Availability

Correspondence and requests for data and materials should be addressed to W.Y.W. (wywang@nwpu.edu.cn).

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
