# Peer review of "The Localized Corrosion and Stress Corrosion Cracking of a 6005A-T6 Extrusion Profile"

_materials, 2021, doi:10.3390/ma14174924_

Round 1

Reviewer 1 Report

Dear authors,

I have read your manuscript with interesting and attention.
My opinion is positive generally, but I have few comment which could help you to improve your paper.

  1. I suggest to avoid three and more citation in one bracket.
  2. I suggest to avoiding three or more quotes in one parenthesis. Otherwise, the citations are too generally related to the editing under discussion.
  3. I suggest to rewrite the conclusions and set them in points for easier reading them by the reader.

Regards

Author Response

Thanks a lot for your suggestions and recommendation of our manuscript.  Please find the response letter in the attachment.  

Reviewer 2 Report

In this manuscript, the localized corrosion characteristics of 6005A-T6 aluminum alloy have been investigated. The authors have used several characterization tools to understand the mechanisms involved. At one side the research seems to be very interesting but on the other side, it could have been managed more professionally. Some comments are given below,

1) Authors are required to carefully go through all the manuscript again as it contains alot of grammatical errors and incomplete sentences. Ex;

Line 62-63: Hydrogen-enhanced localized plasticity (HELP), hydrogen-enhanced vacancy stabilization mecha-62 nism (VM) [11, 19-22].

Line 107-108: Based on the measured crack length at different time, the crack propagation 107 rate (?=????⁄) and the stress intensity factor (K).

Line 156-157: Based on the characterizations and analysis on the crack 156 tips shown in Figure 4(d-f), the pitting and the IGCs.

and several more.

2) Show the directions in Fig. 1a

3) The quality of Fig.2 and 3 is very poor. 

4) Line 172-173 needs more explanation and evidence.

5) There is no Fig. 7 in the manuscript.

6) Explanation of Fig. 8 and 9 can be managed in a more professional way.

7) Discussion section is very poorly written.

The main issue with this manuscript is, even though the study is quite interesting but it is not well managed. There is no symmetry in the paper. It is very difficult for the reader to reach any conclusion. The quality of the figures is very poor. Although, the authors have utilized EBSD yet the explanation is not well written. 

Author Response

(The authors gave the same response as above.)

Reviewer 3 Report

-The issue of the effect of hydrogen-assisted cracking of Al needs to be discussed with more evidence.
-The conclusion part should be rewritten and the most important research achievements should be presented in numerical detail.
-Conditions and corrosion test results in the materials and methods should be provided in more detail.
-The conditions for applying the stress and the dimensions of the sample should be given in more detail.
-More evidence needs to be provided on the formation and effect of the Mg2Si phase on the properties. In this regard, reference to the following sources is emphasized.
-The literature review is not sufficient and authors must review more papers in the field of Al-intermetallic compounds and especially newly published ones. Try to exclude the outdated refs. Doing this, reviewing the following refs could be of help:
[a] Measurement: Journal of the International Measurement Confederation, 77, 2016, 50-53
[b] Materials Science and Engineering A, 657, 2016, 431-440
[c] Materials Research Express 6 (10), 1065d9

Author Response

(The authors gave the same response as above.)

Round 2

Reviewer 2 Report

The manuscript can be accepted in the present form. 

Reviewer 3 Report

The revised manuscript could be published in Materials.